# Impact of Lifestyle Interventions on Multiple Sclerosis: Focus on Adipose Tissue

**DOI:** 10.3390/nu16183100

**Published:** 2024-09-14

**Authors:** Marta Mallardo, Filomena Mazzeo, Giacomo Lus, Elisabetta Signoriello, Aurora Daniele, Ersilia Nigro

**Affiliations:** 1Department of Molecular and Biotechnological Medicine, University of Naples “Federico II”, 80138 Naples, Italy; mallardo@ceinge.unina.it (M.M.); aurora.daniele@unina.it (A.D.); 2CEINGE-Biotechnologies Advances S.c.a r.l., Via G. Salvatore 486, 80145 Naples, Italy; ersilia.nigro@unicampania.it; 3Department of Economics, Law, Cybersecurity and Sports Sciences (DiSEGIM), University of Naples “Parthenope”, 80035 Naples, Italy; 4Multiple Sclerosis Center, II Neurological Clinic, University of Campania “Luigi Vanvitelli”, 80131 Naples, Italy; giacomo.lus@unicampania.it (G.L.); elisabetta.signoriello@unicampania.it (E.S.); 5Department of Medical and Surgical Sciences, University of Campania “Luigi Vanvitelli”, 80131 Naples, Italy; 6Department of Pharmaceutical, Biological, Environmental Sciences and Technologies, University of Campania “Luigi Vanvitelli”, Via G. Vivaldi 42, 81100 Caserta, Italy

**Keywords:** multiple sclerosis, physical activity, adipose tissue, adipokines, nutritional supplements, diet

## Abstract

Multiple sclerosis (MS) is a chronic autoimmune disorder characterized by demyelination in the central nervous system (CNS), affecting individuals globally. The pathological mechanisms underlying MS remain unclear, but current evidence suggests that inflammation and immune dysfunction play a critical role in the pathogenesis of MS disease. Adipose tissue (AT) is a dynamic multifunctional organ involved in various immune diseases, including MS, due to its endocrine function and the secretion of adipokines, which can influence inflammation and immune responses. Physical activity represents an efficacious non-pharmacological strategy for the management of a spectrum of conditions that not only improves inflammatory and immune functions but also directly affects the status and function of AT. Additionally, the exploration of nutritional supplementation represents an important field of MS research aimed at enhancing clinical symptoms and is closely tied to the regulation of metabolic responses, including adipokine secretion. This review, therefore, aims to elucidate the intricate relationship between lifestyle and MS by providing an overview of the latest published data about the involvement of AT and the main adipokines, such as adiponectin, leptin, and tumor necrosis factor α (TNFα) in the pathogenesis of MS. Furthermore, we explore whether physical activity and dietary management could serve as useful strategies to improve the quality of life of MS patients.

## 1. Introduction

Multiple sclerosis (MS) is an autoimmune demyelinating disease of the central nervous system (CNS), currently affecting over 2.5–3 million people worldwide [1,2]. Clinically, MS is a heterogeneous disease classified into three main phenotypes: relapsing–remitting multiple sclerosis (RRMS), primary progressive multiple sclerosis (PPMS), and secondary progressive multiple sclerosis (SPMS). RRMS affects approximately 85% of MS patients and is characterized by episodes of neurological dysfunction alternating with periods of remission. Many RRMS patients eventually transition to SPMS within 20 years, characterized by worsening neurodegenerative processes. Alterations in cytokine levels and costimulatory molecules have been noted in the dendritic cells of patients moving from RRMS to SPMS. Other factors contributing to the death of neuronal and oligodendrocyte cells may involve elevated nitric oxide production and the release of soluble proteins like matrix metalloproteinases. These processes exacerbate oxidative stress, directly affecting the integrity of the blood–brain barrier (BBB) [3]. PPMS, affecting the remaining 10–15% of patients, involves continuous neurological damage from disease onset [3,4]. Despite clinical distinctions, chronic inflammation and immune system dysfunctions play a critical role across all MS phenotypes [5]. While MS was once considered strictly related to T cell dysregulation and mainly driven by autoreactive T helper (Th)1 and Th17 cells, it has now been recognized that numerous factors, including the endocrine activity of adipose tissue (AT), contribute to MS immunopathology [6]. Apart from its role in energy homeostasis, AT functions as an active endocrine organ, significantly influencing immune and inflammatory processes through the release of adipocytokines [7]. Adiponectin, leptin, and tumor necrosis factor α (TNFα) are key adipokines released by AT that significantly influence immune responses and inflammation. Adiponectin typically has anti-inflammatory effects, although its role in MS is complex, with research indicating that it may have both protective and detrimental effects depending on the disease stage. In contrast, leptin is pro-inflammatory and enhances Th1 and Th17 responses, which are critical in MS pathology. TNFα, a well-known pro-inflammatory cytokine, drives the inflammatory response by activating immune cells and fostering demyelination [8,9,10]. Disruptions in the secretion of these adipokines might be involved in the pathophysiology of MS, potentially heightening the risk of disease onset and accelerating its progression, as evidenced by their altered levels in MS patients compared to healthy controls [11,12,13,14].

Physical activity is increasingly recognized as a powerful non-medical tool for treating various diseases, enhancing metabolic, immune, and inflammatory functions [15,16,17]. An increasing number of studies in the literature support the idea that regular physical activity is particularly beneficial for individuals with MS, given its positive effects on relapse rate, brain atrophy, and disability progression. Well-structured exercise programs have been demonstrated to be safe and effective in managing MS, improving muscle function, walking, mobility, and cognitive functions [18]. Moreover, physical activity can significantly enhance mood, reduce fatigue, and improve overall quality of life, which is especially important for MS patients dealing with chronic and progressive symptoms [18,19]. Importantly, in MS patients, regular exercise has been shown to modulate immune responses, potentially reducing pro-inflammatory cytokines and increasing anti-inflammatory ones produced by AT, thereby modulating its endocrine function [17,20]. This modulation may help to mitigate the chronic inflammation that characterizes MS.

In addition to physical activity, several other lifestyle factors, such as diet, have been shown to significantly influence MS disease activity [21,22]. The nutritional status of MS patients has been proposed as a potential factor influencing MS symptoms and progression [22]. Various dietary supplements appear to possess antioxidant and anti-inflammatory properties, potentially enhancing autoimmune tolerance in MS patients [23,24]. Conversely, recent data from the literature have highlighted a consistently high prevalence of deficiencies in vitamins A, B12, and D3 among MS patients, underscoring the importance of adopting a healthy dietary regimen and appropriate supplementation for individuals with MS [23]. Despite extensive reviews about the contributing factors, signs/symptoms, and dysregulated mechanisms of MS, the role of AT in MS has not been comprehensively reviewed. Understanding the interplay between AT and MS, particularly in relation to physical activity, could provide new insights into therapeutic strategies aimed at managing inflammation and improving overall outcomes for MS patients. Moreover, improving the understanding of the impact of nutritional factors on MS pathogenesis could offer a complementary approach to disease management.

Therefore, the primary outcome of this review was to elucidate the complex relationship between lifestyle and MS by focusing on the roles of AT and the main adipokines—adiponectin, leptin, and TNFα—in the pathogenesis of MS.

Furthermore, we investigated whether physical activity and dietary management can improve the quality of life for MS patients, with particular attention paid to the impact of physical activity on the levels of the aforementioned adipokines in MS patients.

## 2. MS and Physical Exercise

The benefits of exercise in improving physical performance, mental function, and general wellbeing are evident. Interestingly, as reviewed by Bonanni R. et al., even moderate exercise can be beneficial in the context of neurodegenerative diseases. It helps improve energy balance, fitness levels, flexibility, and quality of life by reducing neuroinflammation, enhancing neuroplasticity, and supporting myelin repair [25].

For patients with MS, an exercise prescription is essential [26,27,28,29,30,31,32]. Indeed, evidence indicates that a supervised and personalized exercise program can cause important improvements in different areas of cardiorespiratory fitness, muscle strength, flexibility, balance, fatigue, quality of life, and respiratory function in individuals with MS [33,34,35,36]. Regarding the type of exercise, the majority of studies in the literature have included exercise programs involving both resistance (e.g., progressive resistance exercise, walking mechanics) and endurance training (e.g., bicycle ergometry, arm or arm–leg ergometry, aquatic exercise, treadmill walking), as well as combined training approaches [29,31]. These studies have shown that such exercise regimens provide significant benefits for MS patients, improving disease symptoms. Accordingly, several important physiological and functional benefits of exercise could be listed for MS, including improved aerobic capacity [37], better balance [38], improved mood (with potential reduction in depression) [39], increased muscle strength [40], and improvements in the immune system [41] (see Figure 1).

It is worth noting that the effects of exercise on MS have primarily been studied in patients with mild to moderate impairment, typically those with an Expanded Disability Status Scale (EDSS) score of less than 7 [27]. For patients falling within the mild to moderate range, current exercise guidelines recommend a combination of aerobic and resistance training. Specifically, these guidelines suggest 2–3 days per week of aerobic training, consisting of 10–30 min at moderate intensity, alongside 2–3 days per week of resistance training [36]. To our knowledge, only one study has investigated exercise in highly impaired MS patients with an EDSS score between 5 and 8, highlighting the need for more research to understand the benefits and optimal exercise for individuals with more severe MS symptoms [42].

Despite the potential benefits of exercise, only a few MS subjects (20%) engage in regular physical activity [43,44]. Most likely, common clinical features of MS (e.g., spasticity, weakness, fatigue, impaired balance) contribute to low levels of physical activity rather than a lack of interest in exercise.

The molecular mechanisms responsible for the beneficial effects of exercise in MS are multifaceted. Exercise training has been shown to positively impact individuals with MS by influencing various biological processes, including the regulation of cytokine production, potentially contributing to reduced physical disability [45,46]. Accordingly, research indicates that regular exercise can lead to favorable changes in cytokine profiles, promoting a more balanced immune response and potentially mitigating the neuroinflammatory processes characteristic of MS [47].

Furthermore, exercise-induced neurotrophic factors have been implicated in the preservation of neuronal health and function in MS. Studies suggest that exercise may enhance the production of these factors, thereby promoting neuroprotection and neuroplasticity in MS patients [48]. This neuroprotective effect is crucial, as MS is characterized by the progressive loss of neurological function due to demyelination and neurodegeneration [49].

The positive effects of exercise in MS underscore the importance of incorporating regular physical activity into the comprehensive care regimen for MS patients. Ongoing research into these mechanisms enhances our understanding of MS and helps develop personalized exercise programs that optimize outcomes and improve overall health and quality of life for individuals with MS.

## 3. Involvement of AT in MS Pathophysiology and Adipokine Modulation by Exercise

Clinical and experimental data suggest that the pathogenesis of MS is closely linked to immune system dysfunctions and chronic inflammation [1,6]. The induction of inflammation by the immune system may be influenced by various factors, potentially including the endocrine function of AT, though this involvement is not yet fully explored [13,50]. AT is now recognized as a multifunctional dynamic organ involved in numerous physiological and pathological processes, such as energy metabolism regulation, insulin sensitivity, and immune and inflammatory responses [51]. It is traditionally divided into white adipose tissue (WAT), which stores energy as triglycerides in unilocular adipocytes, and brown adipose tissue (BAT), composed of multilocular, mitochondria-rich adipocytes involved in thermogenesis and energy expenditure [52]. Both WAT and BAT are highly metabolically active organs that secrete adipocytokines and batokines, which play critical roles in several pathophysiological processes, including within the CNS [53]. Adipokines have been suggested to be the molecular link between AT and the inflammatory and immunologic activation of the CNS [53]. While adipokines are known to predict progression in other chronic inflammatory diseases, their involvement in MS has been less explored [54]. Recent evidence indicates that an imbalance between pro-inflammatory and anti-inflammatory adipokines may play a role in the immune–pathological processes associated with MS [13,55,56].

Moreover, regular physical activity has been shown to provide a wide range of benefits on immune function and inflammatory responses, potentially mitigating the progression and severity of various immune-mediated diseases such as rheumatoid arthritis, systemic lupus erythematosus, and inflammatory bowel disease [57]. As previously mentioned, exercise may also play a significant role in alleviating physical disability in individuals with MS. This effect is thought to be achieved, in part, through the regulation of adipocytokine production, which helps to modulate the immune response and potentially slow disease progression [47]. Specifically considering adipokines, there are limited data on the effects of exercise on adipokine profiles in MS. However, one of the molecular mechanisms through which physical activity exerts its positive effects on individuals with MS may be through the involvement of AT endocrine function.

Thus, given the potential role of adipokines in MS pathophysiology and the benefit of physical exercise for individuals with MS, researchers have proposed a renewed focus on the effects of exercise on adipokines. This focus is partly supported by evidence that regular exercise induces an anti-inflammatory response in AT and by the broader effects of exercise on immune system markers in the general population [58,59,60]. Exercise positively modulates adipokine release through several mechanisms, primarily by reducing fat mass and stimulating the release of various factors that exert anti-inflammatory effects and influence AT metabolism, promoting the release of adiponectin and other beneficial adipokines [61,62,63]. Thus, by modulating the peripheral immune system, it may be possible to indirectly influence CNS inflammation in MS.

In the following sections, we will delve into the role of key adipokines, such as adiponectin, leptin, and TNFα, in MS pathogenesis. Additionally, we will explore how exercise affects these adipokines and the potential benefits it offers to individuals with MS.

### 3.1. Adiponectin

Adiponectin is a 244-amino acid protein synthesized by adipocytes. It forms low (LMW)-, medium (MMW)-, and high-molecular-weight (HMW) complexes, with HMW oligomers having the most significant biological effects [64]. Adiponectin acts through specific receptors: AdipoR1 (mainly in skeletal muscle), AdipoR2 (predominantly in the liver), and T-cadherin (mainly in the cardiovascular system) [65].

Adiponectin plays a crucial role in regulating insulin sensitivity, glucose and lipid metabolism, and exhibits anti-inflammatory, anti-fibrotic, and antioxidant properties [66]. It also modulates immune response, with its effects varying based on the type of receptor activated on immune cells. Accordingly, changes in adiponectin levels have been reported in numerous immune-related diseases, including inflammatory diseases like inflammatory bowel disease and autoimmune diseases such as systemic lupus erythematosus [67]. Adiponectin’s mechanisms on immune cells are well documented, particularly its anti-inflammatory activity [67,68]. It suppresses the release of pro-inflammatory cytokines TNFα, IL-6, and IL-8 from monocytes while inducing the production of anti-inflammatory mediators IL-10 and IL-1 receptor antagonists [69]. However, pro-inflammatory activities of this adipokine have also been reviewed by Choi H.M. et al. [67].

The role of adiponectin in MS remains controversial. The majority of studies suggest an increase in total serum adiponectin levels in MS patients [67].

For instance, a study on blood samples from 99 MS patients and 89 healthy subjects found higher adiponectin levels in MS patients. Additionally, follow-up over 3.6 ± 2.20 years confirmed the prognostic value of adiponectin, as patients with higher levels had worse EDSS [13]. Similarly, Çoban et al. supported the involvement of adiponectin in both the pathogenesis and progression of MS, suggesting that higher adiponectin levels could serve as prognostic biomarkers for MS [70]. Düzel et al. also found significantly higher levels of adiponectin and other adipokines in RRMS patients compared to healthy controls [71]. High adiponectin levels in MS may indicate a significant attempt by AT to counteract chronic inflammation; this response suggests that AT is actively trying to mitigate the ongoing inflammatory processes in MS, although it may be insufficient to fully counteract the disease’s progression. It is plausible that signaling pathways exist through which the CNS communicates with AT to signal that “help is needed”, although the precise mechanisms still remain unclear. Accordingly, the CNS can directly influence AT activity through neural and neuroendocrine signals, including hormones and neurotransmitters, which regulate adipokine production and release [72]. For example, sympathetic nervous system (SNS) activation via β-adrenergic receptors modulates BAT activity, affecting thermogenesis and energy metabolism [73]. This interaction may have implications for MS, suggesting a role for peripheral neurons and neuroendocrine pathways in systemic inflammation regulation. Moreover, altered signals from the MS-affected brain microenvironment may influence adipokine secretion from AT, thereby impacting systemic inflammation and disease progression [11].

However, some research indicates lower adiponectin levels in MS patients compared to healthy individuals [74]. Moreover, a 2-year randomized controlled trial involving 88 MS patients found no significant differences in adiponectin levels relative to disease severity or treatment response [75]. These discrepancies could be due to the small sample sizes and the inclusion of MS patients undergoing disease-modifying treatments.

The cerebrospinal fluid (CSF) levels of adiponectin may serve as a useful indicator of MS disease, as two studies have also reported higher CSF adiponectin levels in MS patients compared to controls [6,76]. Signoriello et al. found that CSF adiponectin levels are higher in MS patients than in controls, with particularly high levels in primary progressive MS (PPMS) compared to relapsing–remitting MS (RRMS). Elevated CSF adiponectin was associated with higher baseline EDSS scores and more severe disease at a 4.5-year follow-up. Additionally, adiponectin levels correlated with CSF IgG levels and showed an altered oligomerization profile, with significant increases in HMW and MMW isoforms [6]. Although limited by a small sample size, Hietaharju et al. reported that CSF adiponectin of MS patients is significantly higher compared to their asymptomatic co-twins [76]. Additionally, in vitro studies provide further evidence supporting the functional role of adiponectin in MS. Piccio et al. showed that adiponectin knockout mice with experimental autoimmune encephalomyelitis (EAE) display heightened inflammation, demyelination, and axonal damage in the CNS. Conversely, administration of adiponectin ameliorates EAE by enhancing the number of T-regulatory cells [77]. Zhang et al. found that adiponectin possesses properties capable of inhibiting autoimmune inflammation mediated by Th17 cells in the central nervous system in vitro [78]. Collectively, these findings indicate a significant association between adiponectin modulation and disease progression and severity in MS.

The abovementioned evidence underscores the substantial impact of adiponectin on immunological function in MS. Therefore, exploring how adiponectin responds to exercise in an MS population remains a critical yet understudied area in MS research. Considering physical activity as a non-pharmacological intervention in the management of other diseases such as metabolic disorders, it is already well documented that improvements in physiological outcomes are associated with the modulation of adiponectin levels [21]. To date, three studies have examined adiponectin levels following exercise in people with MS, reporting varied outcomes: an increase, a decrease, and unchanged levels of this adipokine [12,79,80]. For instance, an 8-week aerobic interval training study showed improvements in both psychological and physiological parameters, alongside increased adiponectin levels in women with MS, suggesting exercise’s beneficial effects on quality of life and fatigue by influencing adipose tissue function [79]. This enhancement of adiponectin levels in response to exercise may signify an anti-inflammatory effect of exercise training in MS subjects. Conversely, a case report by Grazioli et al. suggested that a well-structured concurrent aerobic and resistance training program reduced adiponectin levels and HMW oligomers within 4 months, with sustained effects observed at the 6-month follow-up [12]. The results also showed a significant improvement in the body composition profile of MS subjects [12]. Another study on RRMS patients found no significant changes in adiponectin levels immediately post-exercise, potentially due to variations in exercise duration, type, or the clinical and metabolic profiles of the MS participants [80]. While these studies focused on serum samples, no data are available on CSF adiponectin levels in MS patients. However, Schön et al. demonstrated that acute intense aerobic exercise modulates numerous cytokines in the CSF of healthy young volunteers, with adiponectin showing the most significant exercise-induced changes [81]. Considering the critical role of inflammation in both the onset and progression of MS, there is a pressing need for further research to investigate how physical exercise, particularly during early and/or relapsing–remitting phases, influences inflammatory markers such as adiponectin in MS. This exploration aims to elucidate whether these changes are causally related or merely coexist as by-products of another underlying mechanism.

### 3.2. Leptin

Leptin is a well-known pro-inflammatory cytokine that plays a crucial role in regulating energy balance, appetite, and metabolism [82]. Beyond its metabolic functions, leptin is heavily involved in immune responses, contributing to the inflammatory state observed in several chronic conditions, including MS [83]. Its pro-inflammatory effects are widespread, influencing both innate and adaptive immunity [83]. Leptin exerts these effects by binding to its receptors, of which six leptin receptor (LepR) isoforms have been identified, each with distinct physiological roles [84]. In human T cells, B cells, and monocytes, activation of these cells leads to a significant increase in LepR expression [85]. Leptin treatment in activated B cells results in enhanced pro-inflammatory cytokine production, including IL-6, TNFα, and IL-10 [86]. Conversely, leptin significantly inhibits the proliferation of regulatory T (Treg) cells, which are crucial for maintaining immune tolerance and preventing autoimmune responses [86].

Leptin enhances the production of pro-inflammatory cytokines such as IL-1, IL-6, IL-12, and TNFα in monocytes [87]. On the other hand, pro-inflammatory cytokines like IL-1β, TNFα, and IL-6 stimulate the secretion of leptin by mononuclear phagocytes [87]. This bidirectional interaction between leptin and pro-inflammatory molecules exacerbates the inflammatory response, potentially significantly contributing to the pathology of MS. Fifteen studies have investigated leptin levels in MS patients, with nine reporting significantly higher serum leptin levels compared to healthy controls [55,71,88,89,90,91,92,93,94,95,96,97,98,99,100]. Six studies found no difference in leptin levels, and one study reported lower levels in MS patients [95,96,100,101,102,103,104,105]. Although no clear association between serum leptin levels and EDSS scores was found in patients with RRMS, positive correlations between serum leptin and EDSS were observed in patients with SPMS and PPMS [105]. Regarding CSF leptin levels, one study reported no significant changes, while another observed significantly higher leptin levels in RRMS patients [76,106,107].

Three trials investigated the impact of regular physical activity on circulating levels of leptin in individuals with MS [79,107,108]. Ebrahimi et al. found no significant changes in leptin levels or body mass index [108]. In this study, the intensity of the exercise is likely to not have been sufficient to achieve maximal benefits on leptin levels. Conversely, Mokhtarzade et al. reported a notable decrease in serum leptin levels alongside improvements in body composition among MS participants [79]. Similarly, Majdinasab et al. also noted a significant post-exercise reduction in leptin levels in individuals experiencing relapses [80].

Although there are only two studies reporting reduced levels of leptin following exercise, the decline in leptin levels in MS suggests a potential anti-inflammatory effect of physical activity. As mentioned above, leptin influences immune responses by modulating Th1 and Th2 cell functions. Lower levels of leptin are typically associated with suppressed Th1 cell activity and enhanced Th2 cell function, which results in reduced production of pro-inflammatory cytokines. Therefore, the observed reduction in leptin levels following exercise may indicate a beneficial modulation of immune responses toward a less inflammatory state in MS patients. Nonetheless, further comprehensive investigations are necessary to fully understand and confirm the potential anti-inflammatory effects of exercise on leptin levels in individuals with MS.

### 3.3. TNFα

TNFα is a versatile cytokine that is part of the TNF receptor ligand superfamily. It participates in various homeostatic and inflammatory processes and is significantly involved in autoimmune and inflammatory disorders [109,110,111]. Beyond its role in regulating the inflammatory response, TNFα also has important pathophysiological functions within the CNS [111,112,113].

Regarding MS disease, TNFα is closely linked to MS-related inflammatory demyelination [114,115]. Although TNFα has been linked to the inflammatory processes in MS, recent studies also highlight its potential neuroprotective effects, such as promoting remyelination by increasing oligodendrocyte proliferation [116,117]. Elevated levels of TNFα have been observed in the CSF of MS patients, correlating with disease progression [118]. Similarly, TNFα expression is upregulated in experimental autoimmune encephalomyelitis (EAE), and administering TNFα to these mice has been shown to exacerbate the disease [119,120]. However, the role of TNFα in MS and EAE is complex and not solely detrimental [119,120]. Surprisingly, research has shown that TNFα knockout mice exhibit EAE symptoms that are as severe, or even more severe, than those observed in wild-type mice [121].

The intricate role of TNFα in MS is likely attributed to the complexity of its signaling pathways and its interactions with various cytokines, CNS cells, and immune cells. TNFα interacts with two distinct receptors: TNFR1 and TNFR2. TNFR1, which is broadly expressed, is involved in tissue degeneration and inflammation, whereas TNFR2, selectively expressed in neurons, microglia, oligodendrocytes, T cells, and endothelial cells, mediates homeostatic functions [122].

The impact of physical activity on TNFα modulation in individuals with MS has been investigated in several trials [79,80,123,124,125,126,127,128,129,130,131]. Some studies found no significant changes in TNFα levels following exercise training [124,125,128,129,130], while others indicated a decrease in serum levels [79,80,123,126,131]. Although Kjølhede, et al. reported no changes in TNFα levels, they observed a reduction in IL-17 secretion following resistance exercise in the trained subjects compared to the untrained ones, supporting the anti-inflammatory effect of physical exercise in individuals with MS [129]. Only one study reported an increase in TNFα levels [127].

These variations in the effects of exercise on TNFα levels may be attributed to its pleiotropic nature. As previously mentioned, TNFα can be linked to both harmful effects on the myelin sheath and BBB, as well as the promotion of remyelination. This dual effect may be attributed to the activation of two distinct signaling pathways mediated by separate TNFα receptors, namely p55 and p75 [116,117]. Exercise might specifically activate the “beneficial” TNFα-p75 receptor pathway, which encourages cell growth and proliferation [116]. These findings underscore the critical role of TNFα in MS. The variations in cytokine production and release may be influenced by factors such as the method of training, timing of exercise, and the type of sample (CSF or blood) used for analysis. Accordingly, further research is required to obtain more comprehensive results regarding TNFα responses to exercise in MS and its associations with disease symptoms.

Table 1 summarizes the main findings on adipocytokine changes in relation to MS.

## 4. Nutrition and Supplements

Health promotion requires good nutrition and an adequate lifestyle. In recent years, an increasing number of scientific studies have demonstrated how an unwholesome diet, typically high in sugar, saturated fat, and salt and low in vegetables and fruits, can participate to the development of non-communicable commune diseases (NCDs), such as cancer, heart syndromes, diabetes, and neurodegenerative disorders such as MS [132]. Indeed, aside from established epidemiologic evidence that smoking, vitamin D, and Epstein–Barr infections impact on the risk of developing MS, there is rising interest in diet and its effects on both the development and progression of this disease [133]. What to eat is an essential factor to consider, as dietary factors are capable of controlling the expression of specific genes and thereby driving metabolic pathways.

Nutrition intervention studies suggest that diet may be contemplated as an integrative treatment to control the progression of the disease [134]. Scientists have not recognized a definitive effective diet that can change the course of MS, but clinical trials suggest a benefit from linoleic acid and from the intake of vitamin D, both associated with a lower incidence of MS [135]. A ketogenic diet (high-fat, low-carbohydrate) has also demonstrated advantageous effects on neurodegenerative diseases through the modulation of central and peripheral metabolism, mitochondrial function, inflammation, oxidative stress, autophagy, and gut microbiome [136]. More specifically in MS, it has been suggested that a ketogenic diet improves clinical symptoms by promoting central inflammation (particularly through an increase in interleukins IL-1β, IL-6, and IL-17), and reducing emotional disorders such as anxiety and depression [137,138]. The overcoming MS (OMS) diet is a plant-based diet (food that has been processed or refined as little as possible) that comprises seafood and fish but cuts out all processed foods, meat, eggs, dairy, and saturated fats. Additional supplements may be needed to ensure patients receive enough nutrients, like protein, iron, and calcium [139]. The OMS diet has been cross-sectionally associated with lower fatigue, depression, and disability in MS patients [140]. Observance to the OMS diet, as part of a multimodal lifestyle program, has also been associated with improved quality of life (QoL) and reduced fatigue and depression among MS patients [141].

Numerous trials involving omega-3 fatty acid supplementation (e.g., fish oils, EPA and DHA acid, 6–10 g per day for 1–2 years) have been lately conducted among patients with MS [142]. Additionally, antioxidant factors such as vitamins and unsaturated fatty acids have been studied and seem to play a role in the regulation of oxidative stress in MS [143].

In the next paragraphs, we will analyze the principal nutrition factors effective in the management of MS patients in relation to plausible modifications in adipokines expression.

Polyunsaturated fatty acids (PUFAs) are extremely effective antioxidant compounds. Bjørnevik et al. reported a low incidence of MS in people following diets enriched in PUFAs [144]. Some studies have shown that PUFAs also act against the progression of the disease, reducing the frequency of relapses [145]. In human studies, a low-fat diet supplemented with PUFAs was linked to reduced disability scores on the EDSS, slight improvements in relapse rates and fatigue, and an overall enhancement of quality of life [143]. Such effects might be associated with the amelioration of neurodegeneration and a consistent decrease in demyelination in MS. The molecular mechanisms of such effects are related to a decrease in inflammation and the maintenance of immunomodulation [143]. Ramirez et al. described the beneficial effects of supplementation with fish oil (comprising high amounts of omega-3 PUFAs) against inflammation and oxidative stress [146]. Omega-3 fatty acid supplementation results in a reduction in pro-inflammatory cytokines and free radicals [146,147,148]. A recent review by Al-Naqeb et al. examines the health benefits of ten plant-based oils—primarily seed oils—on MS. These include pomegranate seed oil, sesame oil, Acer truncatum seed oil, hemp seed oil, evening primrose oil, coconut oil, walnut oil, essential oil from Pterodon emarginatus seeds, flaxseed oil, and olive oil [149]. The authors conclude that plant-based oils may be beneficial in managing MS and its associated symptoms. Their potential benefits include reducing inflammation, promoting remyelination, modulating the immune system, and inhibiting oxidative stress [149].

With regard to adiponectin modulation, to our knowledge, there is not any evidence specifically found in MS patients, but, in diabetes patients, *n*-3 PUFAs supplementation is able to increase AdipoR1 and AdipoR2 gene expression and adiponectin serum levels [150,151]. Similarly, in patients with stable coronary artery disease, omega-3 PUFA supplementation improves the adiponectin profile [152]. Thus, such data suggest that a possible modulation of adiponectin following supplementation with PUFAs in MS might exist.

Vitamins have also been largely studied in MS. Vitamin D plays a significant role not only in calcium homeostasis and bone health but also in immunomodulation and the reduction in oxidative stress [153]. Frequently, MS patients exhibit vitamin D deficiency, also associated with a higher risk of the development and relapse of MS [154,155]. A low vitamin D intake has been associated not only with a high risk of developing MS but also with a worsening of the disease and an increased risk of relapses (relapse, fatigue, and disability) [156]. Some evidence suggests that vitamin D supplementation exerts anti-inflammatory and immunomodulatory effects on MS pathogenesis by inhibiting CD4+ T cell production, potentially reducing the risk of MS and slowing disease progression [143]. However, a recent multi-center intervention study did not demonstrate a significant impact of vitamin D supplementation on the course of MS, raising doubts about its effectiveness [157]. As a result, the therapeutic or preventive role of vitamin D in MS is not widely accepted in clinical guidelines. Also known as retinol, vitamin A is a fat-soluble vitamin present in foods of both animal- and plant-based origin (liver, milk, cheese, green leaves, oil, vegetables, and fruit), with a wide variety of functions in visual ability, skin, and immunity. Vitamin A includes various active forms, including retinoids and carotenoids. The association between the pathogenesis of MS and vitamin A remains unclear, although a lack of correlation between the development of MS and low levels of vitamin A has been defined [158,159]. On the other hand, a correlation between vitamin A and the severity of some disease symptoms has also been suggested: a randomized controlled trial presented benefits in terms of fatigue, depression, and cognitive status of MS patients supplemented with high doses of vitamin A [160].

Phytic acid or phytate is the principal reservoir of phosphorus present in almost all wholegrains, legumes, and oilseeds [161]. When phytate is consumed in large amounts by itself without being processed/cooked, it can decrease the absorption of some minerals, leading to the definition of phytate as an antinutrient [162].

The effectiveness of high-phytate foods has been proven to improve cardiovascular health through the molecular mechanisms linked to their ability to prevent vascular calcifications [163].

In neurodegenerative diseases such as Parkinson’s disease, phytate can display strong antioxidation and anti-inflammatory action, blocking the formation of oxygen radicals (OH-) [163,164], inhibiting lipid peroxidation [165] and mitigating neuronal damage and loss [166]. An association between phytate intake and the inhibition of cognitive decline has also been found. Indeed, in MS risk, the assumption of phytate has been found to be involved. The intake of grain or meat, fat, and milk from animals (with a high content of phytic acid) correlated positively with the prevalence of MS [166]. To date, the molecular mechanisms underlying such observations are unknown; however, the ability of phytic acid to influence the bioavailability of several metabolites, such as calcium and vitamin D, might be involved. Behind these effects, phytate can possibly exercise its positive effects too by decreasing leptin and increasing adiponectin levels [167]. Specifically, in diabetic patients, an 8-week diet rich in legumes is significantly able to increase serum adiponectin concentrations [168]. The same patients receiving phytate supplementation showed a significant decrease in serum levels of HbA1c and an increase in adiponectin levels [168]. In particular, InsP6 intake induces an increase in plasma adiponectin concentration in patients with diabetes, indicating that a phytate-rich diet could help to prevent or minimize diabetic-related complications [168].

In terms of probiotics, the latest research has revealed an association between the gut microbiota and the central nervous system as the gut–brain axis, which encompasses a communication network between the nervous, endocrine, and immune systems [169]. In addition, studies have advised that variations in the gut microbiota can significantly impact the inflammatory responses of individuals with MS [170]. Notably, supplementation with probiotics can positively influence both immune and inflammatory responses by reducing serum inflammatory cytokines such as C-reactive protein (CRP), TNFα, and interferon gamma (IFN-γ) [171]. Indeed, probiotic supplementation has led to significant improvements in the disability scores and mental health parameters (reduced depressive symptoms, anxiety, and stress) of MS patients in a 12-week randomized controlled trial. Additionally, it was found to decrease inflammatory markers, including high-sensitivity C-reactive protein (hs-CRP), and oxidative stress markers, such as plasma nitric oxide (NO) metabolites and malondialdehyde (MDA). Probiotic supplementation also enhanced insulin resistance and cholesterol levels [172]. These findings advise that probiotics could play a favorable role for various aspects of MS, including disability, mental health, inflammation, and metabolic conditions.

Table 2 summarizes the main findings on dietary approach and/or supplement in relation to MS.

## 5. Conclusions

MS is a chronic autoimmune disease affecting the CNS over a long-term period. The severity of MS varies widely and is influenced by several factors, comprising the endocrine activity of AT. The involvement of AT in MS pathogenesis remains to be further clarified; certainly, MS is characterized by an altered immune response in which AT also participates through dysregulated adipokine secretion, increasing the risk of disease development and accelerating its progression. Based on the data presented in this review, it is plausible that the functionality of AT is positively influenced by lifestyle factors such as physical activity and nutrition, which are essential in the management of MS.

Currently, physical activity is recognized as a successful rehabilitation strategy for MS patients. Organized and adapted exercise programs can enhance fitness, functional capacity, and overall quality of life, helping as an adjunctive therapy. Additionally, nutritional supplementation represents an important field of MS research aimed at improving MS clinical symptoms. It seems clear that both physical activity and nutrition can be important instruments in the inhibition of MS establishment.

More importantly, once MS is initiated, engaging in regular physical activity and appropriate nutrition can contribute to decreases in disease severity by stimulating the inflammatory response. Therefore, an active lifestyle can be considered an essential part of MS prevention and treatment. Changes in adipokine levels might actively participate in driving, at least in part, the disease-beneficial effects of exercise and nutrition in MS.

In conclusion, the complex interplay between MS pathogenesis, AT endocrine function, and lifestyle factors represents a significant area of research for the prevention and management of the disease. Thus, studies involving larger patient cohorts are necessary to better understand the molecular mechanisms underlying AT’s endocrine response to exercise and nutrition in MS. Such comprehension will represent a step towards the identification of potential novel targets involved in both the establishment and progression of the disease.

## Figures and Tables

**Figure 1 nutrients-16-03100-f001:**
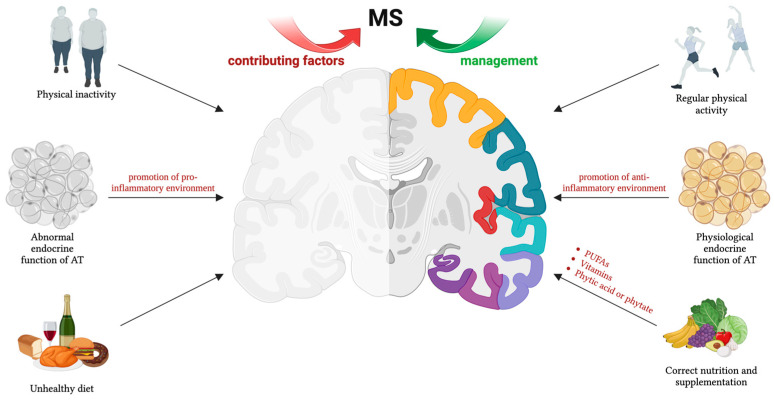
Physical activity, nutrition, and supplementation are crucial non-pharmacological tools for MS management. Lifestyle factors influence AT endocrine function, particularly through adipokine secretion. These hormones play a role in MS pathophysiology, affecting disease development and progression. Thus, maintaining balanced adipokine levels is essential for fostering an anti-inflammatory environment. The figure was created using BioRender.com.

**Table 1 nutrients-16-03100-t001:** Modulation of adipokines in MS in relation to physical exercise.

Adipocytokine	Study Population	Main Findings	Reference
Adiponectin	Case report: a 39-year-old RRMS patient	Total serum adiponectin and HMW oligomers were reduced after 4 months of training at moderate intensity (65% heart rate reserve); in addition, a reduction in BMI (−0.9%) and FAT (−2.6%) and an improvement in the disability level were also demonstrated	[12]
	40 MS women randomized divided into either a non-exercising control or training group	Blood adiponectin levels considerably increased in the training group (8 weeks of aerobic interval training). In addition, the aerobic interval training was associated with improvements in fatigue, quality of life, and maximal oxygen consumption	[79]
	30 MS women and 15 healthy controls	Adiponectin showed no significant difference between non-exercising and training group (a single bout of aerobic exercise at 60–70% maximal heart rate)	[80]
Leptin	30 MS women and 15 healthy controls	Participants performed a single bout of aerobic exercise at 60–70% maximal heart rate. Immediately following exercise, leptin levels significantly decreased in MS subjects	[80]
	34 MS patients with mild to moderate disability randomly divided into a training group (n = 17) and a control group (n = 17)	Non-significantly changed the serum levels of leptin, ghrelin, ghrelin/leptin ratio, testosterone, and testosterone/leptin ratio between no exercise and training subjects (low-intensity exercise three times a week for 10 weeks)	[108]
TNFα	40 MS women randomized into either a non-exercising control or training group.	TNFα levels significantly decreased subsequent to the aerobic interval training (8 weeks of aerobic interval training)	[79]
	30 MS women and 15 healthy controls	TNFα levels were significantly decreased immediately after exercise (a single bout of aerobic exercise at 60–70% maximal heart rate)	[80]
	8 MS patients with low disability	Decrease in fatigue at the end of physical activity intervention (12-week series of combining Pilates and aerobic exercises) accompanied by a significant reduction in TNFα	[123]
	A randomized controlled clinical trial in 60 MS patients	In response to cardiopulmonary exercise test (30 min training at 60% of VO_2_max), TNFα levels stayed unchanged.	[124]
	15 MS women and 10 healthy women.	Blood samples were taken at baseline. TNFα remained unchanged immediately after exercise and two hours after exercise [15 min treadmill (~50% VO_2_ peak)]	[125]
	67 MS patients	Decrease in the production of TNFα at the end of the exercise program (12-week combined exercise)	[126]
	11 MS and 11 non-MS control subjects (8 women and 3 men in both groups)	TNFα increased in MS compared with controls after exercise (30 min of cycle ergometry at 60% of peak O(2) uptake, 3 day/wk for 8 wk at weeks)	[127]
	10 MS female patients	Participants completed 8-week program of twice-weekly progressive resistance training. After training, TNFα showed non-significant reductions	[128]
	35 MS people treated with interferon (IFN)-β	No changes were observed in TNFα levels after a 24-week progressive resistance training respect to a control group	[129]
	15 MS patients and 13 in control group. Twenty healthy controls	TNFα levels were slightly inducible in MS patients completing an eight-week aerobic training program	[130]
	20 subjects (n = 10 MS patients and n = 10 controls)	Serum concentration of the TNFα decreased significantly after a single bout and 6 weeks of aerobic exercise training in the intervention group	[131]

**Table 2 nutrients-16-03100-t002:** Main findings on dietary approach and/or supplement in MS.

Dietary Approach and/or Supplement	Study Population	Main Findings	Reference
Adherence to the ketogenic diet	99 MS subjects	Amelioration of fatigue and depression accompanied by weight loss and reduction in pro-inflammatory cytokines	[136]
Adherence to the OMS diet	Data from an international population of MS followed over 7.5 years	Lower depression rate	[140]
High intake of grain or meat, fat, and milk from animals (elevate content of phytic acid)	75 MS women and 75 healthy controls	Positive correlation with the prevalence of MS	[166]
Omega-3 fatty acid and fish oils supplementation	Systematic review of 5554 studies	Beneficial effects on reducing relapsing rate, inflammatory markers, and improving quality of life	[142]
Diets enriched in PUFAs	80,920 women from Nurses’ Health Study and 94,511 women from Nurses’ Health Study II	Lower incidence of MS. Among the specific types of PUFA, only α-linolenic acid was inversely associated with MS risk	[144]
Omega-3 PUFAs supplementation	10 MS patients	Improvement in quality of life by decreasing relapse rates	[148]
Vitamin D deficiency	92,253 women from the Nurses’ Health Study (NHS) and 95,310 women from the Nurses’ Health Study II (NHS II)	Higher risk of MS incidence	[155]
Low vitamin D intake	Review of literature data	Increased incidence of MS, but the risk–benefit profile of dosage and duration or supplementation needs to be clarified	[156]
Vitamin D supplementation	172 MS patients were randomized: low-dose vitamin D3–high-dose vitamin D3	Lack of significant effects	[157]
Low levels of vitamin A	31 MS patients and 29 matched controls	Lack of correlation with the incidence of MS	[159]
Vitamin A supplementation	101 MS patients in a placebo randomized clinical trial	Significant improvement in fatigue and depression. Improvement also in psychiatric outcomes during interferon therapy	[160]
Probiotics supplementation	40 MS patients	Significant improvement in inflammatory markers, oxidative stress indicators, pain, fatigue, and quality of life	[171]
Probiotics supplementation	60 MS patients	Significant improvements in disability scores and mental health parameters, such as reduced depressive symptoms, anxiety, and stress	[172]

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
