# Peer review of "Impact of Lifestyle Interventions on Multiple Sclerosis: Focus on Adipose Tissue"

_nutrients, 2024, doi:10.3390/nu16183100_

Round 1

Reviewer 1 Report (Previous Reviewer 2)

Comments and Suggestions for Authors

As compared to the previous version the paper was much improved. The authors have delt with most of the inappropriate citation issues. 

However, in my opinion the initial sections (up to line 284 - before chapter 3.1 Adiponectin) are extremely rich in general statements that are supported mostly by non-systemathic reviews. The authors do not go to the studies that support the reviews they cite as to form an original/personal idea and content to a "I heard that..." approach, thus creating a high chance that the meaning of the original research is perverted by "mouth to ear" transmission. I found the exercise-connected part shallow and general, with some exceptions.

Lines 285-616 are much better written and interesting. 

In my opinion the first part of the article may be either entirely re-done, or it could be thansformed in a much shorter introductory part for the Adiponectin-Leptin-TNFa-vitamin D part. 

Regretfully, I do not think that reviewing reviews can create an "important contribution". They do have their own merit in undrestanding and directing our interests and attention, but research studies should be the main origin of the ideas that are synthesized by such a paper. 

In the file I have added some of the individual comments. 

Author Response

As compared to the previous version the paper was much improved. The authors have delt with most of the inappropriate citation issues. 

However, in my opinion the initial sections (up to line 284 - before chapter 3.1 Adiponectin) are extremely rich in general statements that are supported mostly by non-systemathic reviews. The authors do not go to the studies that support the reviews they cite as to form an original/personal idea and content to a "I heard that..." approach, thus creating a high chance that the meaning of the original research is perverted by "mouth to ear" transmission. I found the exercise-connected part shallow and general, with some exceptions.

We sincerely apologize for the inaccuracy in the references. Accordingly, we have revised the references in the section up to line 284 by reducing citations of review articles and including more original research papers.

Lines 285-616 are much better written and interesting. 

In my opinion the first part of the article may be either entirely re-done, or it could be transformed in a much shorter introductory part for the Adiponectin-Leptin-TNFa-vitamin D part. 

Accordingly, we have revised the introductory section and shortening it. However, according to the revision of reviewer 1, a brief comment about the contribution of oxidative stress in MS has been added.

Regretfully, I do not think that reviewing reviews can create an "important contribution". They do have their own merit in understanding and directing our interests and attention, but research studies should be the main origin of the ideas that are synthesized by such a paper. 

In the file I have added some of the individual comments. 

We thank the reviewer for his/her precious work in revising the manuscript. Accordingly, we have addressed all the comments in the attached file and reduced the citations of reviews by including more original articles.

As compared to the previous version the paper was much improved. The authors have delt with most of the inappropriate citation issues. 

However, in my opinion the initial sections (up to line 284 - before chapter 3.1 Adiponectin) are extremely rich in general statements that are supported mostly by non-systemathic reviews. The authors do not go to the studies that support the reviews they cite as to form an original/personal idea and content to a "I heard that..." approach, thus creating a high chance that the meaning of the original research is perverted by "mouth to ear" transmission. I found the exercise-connected part shallow and general, with some exceptions.

We sincerely apologize for the inaccuracy in the references. Accordingly, we have revised the references in the section up to line 284 by reducing citations of review articles and including more original research papers.

Lines 285-616 are much better written and interesting. 

In my opinion the first part of the article may be either entirely re-done, or it could be transformed in a much shorter introductory part for the Adiponectin-Leptin-TNFa-vitamin D part. 

Accordingly, we have revised the introductory section and shortening it. However, according to the revision of reviewer 1, a brief comment about the contribution of oxidative stress in MS has been added.

Regretfully, I do not think that reviewing reviews can create an "important contribution". They do have their own merit in understanding and directing our interests and attention, but research studies should be the main origin of the ideas that are synthesized by such a paper. 

In the file I have added some of the individual comments. 

We thank the reviewer for his/her precious work in revising the manuscript. Accordingly, we have addressed all the comments in the attached file and reduced the citations of reviews by including more original articles.

Reviewer 2 Report (New Reviewer)

Comments and Suggestions for Authors

In their manuscript, Marta Mallardo et collaborators propose a review on physical activity interaction with multiple sclerosis (MS) progression with a special focus on adipose tissue (AT).

Text is composed of 5 sections, contains 2 figures and one table and cites 171 references.

The manuscript deal about a subject of interest, is readable and arise from a group which expertise is well-established.

That said, I have few points that could be considered.

The importance of stressors such as oxidative stress and oxidative damages in MS progression may be discussed by the Authors.

A table summarizing information of “nutrition and supplement” section could be included. 

It could be interesting to discriminate impact of physical activity/nutrition on MS progression depending on MS type and EDSS.

Some typos:

Line 119 sentence starting “For patients with MS is essential an exercise …”should be rewritten.

Line 504 table 1, the word “regarding” (first line) should be corrected.

In table 1 for TNFalpha line, authors wrote “IL-22”.

Author Response

In their manuscript, Marta Mallardo et collaborators propose a review on physical activity interaction with multiple sclerosis (MS) progression with a special focus on adipose tissue (AT).

Text is composed of 5 sections, contains 2 figures and one table and cites 171 references.

The manuscript deal about a subject of interest, is readable and arise from a group which expertise is well-established.

That said, I have few points that could be considered.

The importance of stressors such as oxidative stress and oxidative damages in MS progression may be discussed by the Authors.

We thank the reviewer for his/her comment. We agree that outline the importance of stressors in MS is useful in the context of PA and supplements. Accordingly, in the introduction section, this point is now discussed.

A table summarizing information of “nutrition and supplement” section could be included. 

We thank the reviewer for the useful observation. Accordingly, we have included a table summarizing the role of diet and supplements in MS.

It could be interesting to discriminate impact of physical activity/nutrition on MS progression depending on MS type and EDSS.

The reviewer has outlined a relevant point. Literature data on the effects of physical activity/nutrition on MS progression depending on MS type and EDSS are contrasting, with some papers reporting a correlation with disease parameters while others reporting an absence (for review doi: 10.1177/1756285611430719; doi: 10.1007/s00415-021-10935-6; doi: 10.1093/advances/nmaa063).  Additionally, the reviewer 1 commented: “ I would leave out the part with "and symptoms were rated on the EDSS". First, EDSS is not a really sensitive scale to small changes, and secondly, everything in MS is related to the EDSS, so that doesn't really say anything.”

Therefore, we have reported data on MS severity and progression with caution throughout the manuscript.

Some typos:

Line 119 sentence starting “For patients with MS is essential an exercise …”should be rewritten.

DONE

Line 504 table 1, the word “regarding” (first line) should be corrected.

DONE

In table 1 for TNF alpha line, authors wrote “IL-22”.

DONE

Round 2

Reviewer 1 Report (Previous Reviewer 2)

Comments and Suggestions for Authors

Authors have addresed many of the issues I have previously raised, and have improved the paper, increasing its interest and readability. 

Reviewer 2 Report (New Reviewer)

Comments and Suggestions for Authors

In their revised version of their interesting MS, authors Marta Mallardo et collaborators, correctly address points I raised.

This manuscript is a resubmission of an earlier submission. The following is a list of the peer review reports and author responses from that submission.

Round 1

Reviewer 1 Report

Comments and Suggestions for Authors

This is a review article on exercise and diet in multiple sclerosis, with special focus on adipose tissue and its associated proteins. The thema is an appropriate topic for publication in this journal.

It is well written throughout, but there is a lot of redundancy. There is repetition of the same sentence. I have not checked everything, but there are a number of references that are not cited correctly, so I suggest to check and correct them in full and resubmit? 

EXAMPLES.

citation 6: No mention of adipose tissue.

P6 citation 6, 74-76 -> probably 7, 75-77

P5 just before 3.1

There is a repetition of the sentence.

Finally, we review the impact of exercise on these adipokines and 

its potential benefits for MS progression. 

Besides, there seems no paragraph corresponding to this sentence.

Reviewer 2 Report

Comments and Suggestions for Authors

All citations should be reviewed and checked for consistency, especially if AI was used. I have found multiple instances of incorrect citation, with dicrepancies between the ideea in the text and the substance of the references. For me this suggests an incorrect method of finding relevant sources, with the text being mostly the opinion of the authors and not a result of the reviewed sources. Non-MS related references/ citations/ statements should be clearly marked as so ("in normal volunteers" or "in mice"). In many parts of the article they are mixed and not easily discernable, with the general effect that conclusions that are not MS related are attributed to MS. 

Most of the references are non-systematic reviews and general works. This article is supposed to review scientific data and not personal opinions/data that has already been biased. I agree that reviews are a source of inspiration, but the source articles of those papers should heve been evaluated and cited, not the reviews themselves.

As the most interesting example, reference [67] - "de Sá, L. F.; Wermelinger, T. T.; Ribeiro, E.daS.; Gravina, G.deA.; Fernandes, K. V.; Xavier-Filho, J.; Venancio, T. M.; Rezende, G. L.; & Oliveira, A. E. (). Effects of Phaseolus vulgaris (Fabaceae) seed coat on the embryonic and larval development of the cowpea weevil Callosobruchus maculatus (Coleoptera: Bruchidae). Journal of insect physiology. 2014, 60, 50–57" - is used to support the next statement: "Adiponectin's mechanisms on immune cells are well-documented, particularly its anti-inflammatory activity" or "Adiponectin plays a crucial role in regulating insulin sensitivity, glucose and lipid metabolism, and exhibits anti-inflammatory, anti-fibrotic, and antioxidant properties", depending how one re-numbers the references. The term "adiponectin" does not appear in the article, and I found no connection with what the authors state in either phrase. 

The authors explore mainly the autoimmune mechanisms, and do not go more in depth on the direct effects of adipokines in the CNS (on neurons and glia). They also focus on the peripheral mechanisms and on clinical part while repeatedly bringing forward "MS pathogenesis". Research in MS is abundant, and in my opinion this review could either thouroughly explore mechanisms and properly go into fundamental research, either adhere only to the clinical part and mute these aspects. 

The authors should also be more careful with how they expand the conclusions of references. To exemplify, the third paragraph on page 12 uses a Cochrane review to support "low incidence of MS" in high PUFA regimens. Although I haven't read (yet) that article in its totality, the conclusions of that review mostly express doubt regarding the support of existing litterature for effects on MS evolution (with different parameters), and mention no support for lower incidence. If the Authors really wanted to bring out their own ideea, they should just picked one of the references of the said article that supports their opinion instead of quoting the  entire review.

In MS, prevention of the disease, reduction of disease activity, and sometimes prevention of disease progression have very different meanings and are by no means general terms that can be used interchangeably, as are not MRI activity and clinical activity. Please verify the translation from sources to the review. 

More detailed objections are to be found in the commented file I have attached. 
